# Seasonal shift in timing of vernalization as an adaptation to extreme winter

Susan Duncan[1], Svante Holm[2], Julia Questa[1], Judith Irwin[1], Alastair Grant[3], Caroline Dean[1]*

[1]John Innes Centre, Norwich, United Kingdom; [2]Mid Sweden University, Sundsvall, Sweden; [3]Department of Environmental Sciences, University of East Anglia, Norwich, United Kingdom

**Abstract** The requirement for vernalization, a need for prolonged cold to trigger flowering, aligns reproductive development with favorable spring conditions. In *Arabidopsis thaliana* vernalization depends on the cold-induced epigenetic silencing of the floral repressor locus *FLC*. Extensive natural variation in vernalization response is associated with *A. thaliana* accessions collected from different geographical regions. Here, we analyse natural variation for vernalization temperature requirement in accessions, including those from the northern limit of the *A. thaliana* range. Vernalization required temperatures above 0°C and was still relatively effective at 14°C in all the accessions. The different accessions had characteristic vernalization temperature profiles. One Northern Swedish accession showed maximum vernalization at 8°C, both at the level of flowering time and *FLC* chromatin silencing. Historical temperature records predicted all accessions would vernalize in autumn in N. Sweden, a prediction we validated in field transplantation experiments. The vernalization response of the different accessions was monitored over three intervals in the field and found to match that when the average field temperature was given as a constant condition. The vernalization temperature range of 0–14°C meant all accessions fully vernalized before snowfall in N. Sweden. These findings have important implications for understanding the molecular basis of adaptation and for predicting the consequences of climate change on flowering time.

*For correspondence: caroline.
dean@jic.ac.uk

**Competing interests:** The authors declare that no competing interests exist.

**Reviewing editor**: Detlef Weigel, Max Planck Institute for Developmental Biology, Germany

## Introduction

The sessile nature of plants necessitates that they modulate most aspects of their growth and development in response to external conditions. One aspect of this is the alignment of developmental transitions with seasonal cues. A major seasonal cue is temperature and plants have evolved the ability to integrate daily fluctuations in external temperature in order to monitor long-term trends (*Aikawa et al., 2010*). Exposure to weeks of cold temperature accelerates the transition to flowering in a process called vernalization. In *Arabidopsis thaliana* vernalization involves the quantitative epigenetic silencing of *FLC* (*Michaels and Amasino, 1999*; *Sheldon et al., 1999*). Cold exposure promotes a cell-autonomous epigenetic switch at *FLC* in an increasing proportion of cells (*Angel et al., 2011*, *2015*). This epigenetic switching mechanism requires a Polycomb complex associated with PHD proteins (*De Lucia et al., 2008*), including the cold-induced VIN3 (*Sung and Amasino, 2004*). This enables activation of *FT*, is a potent activator of flowering in *A. thaliana* (*Searle et al., 2006*). At a standard vernalization temperature of 5°C, the length of cold required to achieve complete epigenetic silencing varies between *A. thaliana* accessions and this maps to non-coding cis polymorphisms in *FLC* (*Coustham et al., 2012*; *Li et al., 2014*). Accessions collected from northerly latitudes typically require longer vernalization, for example, the accession Lov-1 originates from near the northerly limit of the *Arabidopsis* range in Lövvik, North Sweden (62.5°N) and requires three months of vernalization to fully accelerate flowering (*Shindo et al., 2006*; *Coustham et al., 2012*).

**eLife digest** Plants are not able to move around and so they need to be able to adapt their growth and development to seasonal changes in their environment. For example, prolonged exposure to cold temperatures during winter can prime some plants to flower when temperatures increase in the spring—a process called vernalization. In these plants, extended periods of cold temperatures lead to lower activity of a gene called *FLC*, which normally inhibits flowering.

In the plant *Arabidopsis thaliana*, vernalization requires several months of exposure to temperatures between 0–6°C. Recently, *A. thaliana* plants from southern Europe were found to vary in the temperature requirements for vernalization, responding to temperatures higher than 6°C. This suggested that plants from northern Europe might vernalize preferentially at lower temperatures. Here, Duncan et al. compared vernalization in a collection of *A. thaliana* plants (or 'accessions') sampled from different regions of Sweden and the UK.

The experiments show that all the accessions needed temperatures above 0°C to vernalize and that vernalization still worked relatively well at temperatures as high as 14°C. The optimal temperature range for vernalization differed between the accessions, but plants from more northern areas did not necessarily vernalize at lower temperatures. For example, for one particular accession from northern Sweden, the temperature that is optimum for vernalization was 8°C, a notably higher temperature than expected.

Historical local climate records suggested that this accession would vernalize before the first snowfall of the winter in North Sweden. Duncan et al. confirmed this proposal with field experiments. Plants were grown in natural field sites in September and then moved into a greenhouse. The experiments show that the plants complete vernalization by November, which strongly suggests that FLC is silenced during autumn rather than during winter, as previously thought. This changed temperature response is due, in part, to a small number of tiny genetic differences in regions of the *FLC* gene that do not code for protein.

These findings have important implications for future studies of vernalization and flowering time, and for understanding how plants will adapt to on going and future climate change. The next step is to understand what causes these changed temperature responses at a molecular level, which should enable selective breeding for flowering and harvest date in a range of crops.

Effective temperature ranges for vernalization have been determined empirically for different plant species, many of which have been incorporated into chilling unit models that are widely used in agriculture (*Byrne and Bacon, 1992*). A genetically informed photothermal model for flowering in *A. thaliana* has assumed that vernalization occurs when daytime hourly temperatures are higher than 0°C and lower than 6°C (*Wilczek et al., 2009*; *Chew et al., 2012*, *2014*). However, accessions from southern Europe have been found to vernalize at constant temperatures significantly higher than 6°C (*Wollenberg and Amasino, 2012*), suggesting that accessions from northerly latitudes might vernalize most efficiently at relatively low temperatures. Here, we show this is not the case and find that vernalization in a range of *A. thaliana* accessions is most effective across a relatively high temperature range with the N. Swedish accession Lov-1, showing maximal vernalization at 8°C. We show that vernalization is complete before snowfall in N. Sweden with the plants flowering immediately upon snowmelt. Vernalization responsiveness in the field matched that when the average monthly temperature was given as constant conditions. Our work has important implications for modeling flowering time and predicting the impact of climate change.

## Results and discussion

In order to investigate natural variation for vernalization temperature requirement in *A. thaliana* accessions we selected several genotypes that represent most of the major *FLC* haplotypes (*Li et al., 2014*): Lov-1 (Lövvik, N. Sweden—latitude 62.5°N), Var2-6 (Vårhallen, S. Sweden—latitude 55.58°N), Ull2-5 (Ullstorp, S. Sweden—latitude 56.06°N), Edi-0 (Edinburgh, UK—latitude 55.95°N) and the reference Columbia line containing *FRIGIDA* (Col *FRI*^Sf2, [*Michaels and Amasino, 1999*]) (*Figure 1—figure supplement 1*). All genotypes were vernalized for varying periods at different constant temperatures between 0°C and 14°C and the efficiency of vernalization assayed by

determining flowering time (*Figure 1A-E*). All the genotypes showed limited vernalization after 4 and 6 weeks exposure to 0°C and vernalized more efficiently at all other temperatures. Col *FRI*$^{Sf2}$ and Edi-0 were most effectively vernalized after 4, 6 or 12 weeks at 2°C, 5°C and 8°C and still vernalized relatively efficiently at 12°C and 14°C (*Figure 1A and C*). Even after 2 weeks of cold at 2°C and 8°C the flowering of Col *FRI*$^{Sf2}$ plants was similar, so lack of any difference was not due to vernalization being close to saturation (*Figure 1—figure supplement 2*). Ull2-5 showed similar temperature sensitivity to Col *FRI*$^{Sf2}$ and Edi-0, but required 12 weeks for vernalization to be fully effective (*Figure 1D*). In contrast, Lov-1 and Var2-6 plants showed a differential response to temperature with 2 and 12°C less effective than 5 and 8°C after 6-weeks vernalization (*Figure 1B and E*). For Lov-1 the only temperature that resulted in flowering after 4 weeks exposure was 8°C and although the enhanced effect of this temperature diminished over time, 8°C consistently resulted in the most effective vernalization (*Figure 1B*). Thus, the different accessions show distinct temperature profiles for vernalization and all require temperatures higher than 0°C.

The requirement for longer cold for effective vernalization in the Lov-1 accession has previously been shown to involve quantitative variation in accumulation of epigenetic silencing of *FLC* (*Shindo et al., 2006*; *Coustham et al., 2012*). We compared this quantitative variation in the silencing of Col *FRI*$^{Sf2}$ and Lov-1 *FLC* alleles after 4 weeks cold exposure at 2, 5, 8, 12 and 14°C (*Figure 2—figure supplement 1A and B*). In contrast to Col *FRI*$^{Sf2}$, the Lov-1 allele re-activated after 30 days in the warm after vernalization at all the tested temperatures. However, the degree of re-activation was lowest after vernalization at 8°C, consistent with vernalization being most effective at this temperature. Similarly, 6 weeks vernalization at 8°C resulted in lower *FLC* re-activation post-cold and higher levels of *FT* induction than 5°C, with similar *VIN3* expression (*Figure 2A–C*). Epigenetic silencing of *FLC* is associated with Polycomb silencing and accumulation of H3K27me3 over the gene body (*Angel et al., 2011*; *Yang et al., 2014*). In Lov-1 it takes longer to accumulate the H3K27me3, mainly due to lower starting levels (*Coustham et al., 2012*). We found similar accumulation of gene body H3K27me3 in the Col *FRI*$^{Sf2}$ *FLC* allele at 5, 8 or 14°C, but differential H3K27me3 accumulation in the Lov-1 allele (*Figure 2D*, *Figure 2—source data 1*). Vernalization at 8°C resulted in higher levels of H3K27me3 compared to 5 or 14°C (*Figure 2D*), suggesting that the Polycomb silencing is most effective at 8°C for the Lov-1 *FLC* allele.

The relatively high temperature range for vernalization of the N. Swedish accession was surprising given that flower buds appeared within 2 weeks of the snowmelt on native *Arabidopsis* at the N. Swedish Lövvik site (*Figure 3—figure supplement 1*). This early flowering may limit herbivory and help in the competition for nutrients (*Kawagoe and Kudoh, 2010*; *Akiyama and Agren, 2012*). A long-term (>5 year study) of the natural populations at several N. Swedish sites throughout the High Coast region, showed most populations behaved as winter annuals with germination occurring predominantly in August and September with no spring germination (*Figure 3—source data 1*). The rapid flowering after snowmelt suggests that vernalization must have occurred before the end of November given the recurrent snow cover and low temperatures at the Lövvik site over the winter months (*Figure 3—figure supplement 2*, *Figure 3—figure supplement 3A*). Hourly climate data collected near Lövvik between 1st August (the earliest germination date observed for natural populations) until snow cover between 2008 and 2013 show an average air temperature of ~8°C (*Figure 3—figure supplement 3A*). Analysis of national data (1st August—30th November) also revealed an overall average autumn daily average temperature of 8.86°C between 1961 and 2008 (SD = 0.63) with over 86% of days falling within the range identified as being effective for Lov-1 vernalization, (0°C, 15°C) (*Figure 3—figure supplement 3B*). The agreement of average autumn temperatures with the effective vernalization temperatures identified for the Lov-1 reinforced the view that epigenetic silencing of *FLC* would occur before snowfall.

We tested the hypothesis of a seasonal shift in the timing of vernalization in N. Sweden by setting up field experiments close to the Lövvik site in autumn 2011 and 2012 (locations shown in *Figure 3—figure supplement 4*). Seedlings were transplanted into the field at the beginning of September and then transferred to a warmed greenhouse at three time points during autumn (*Figure 3A*, *Figure 3—figure supplement 5A*). This enabled us to explicitly test whether 12 weeks of growth preceding winter would be sufficient to fully vernalize Lov-1. Flowering time of the different cohorts showed that vernalization was complete by the end of November in both 2011 (*Figure 3B*) and 2012 (*Figure 3—figure supplement 5B*). Furthermore, plants left to overwinter in the field flowered rapidly at snowmelt, at the same time as the native *A. thaliana* population (*Figure 3—figure supplement 6*).

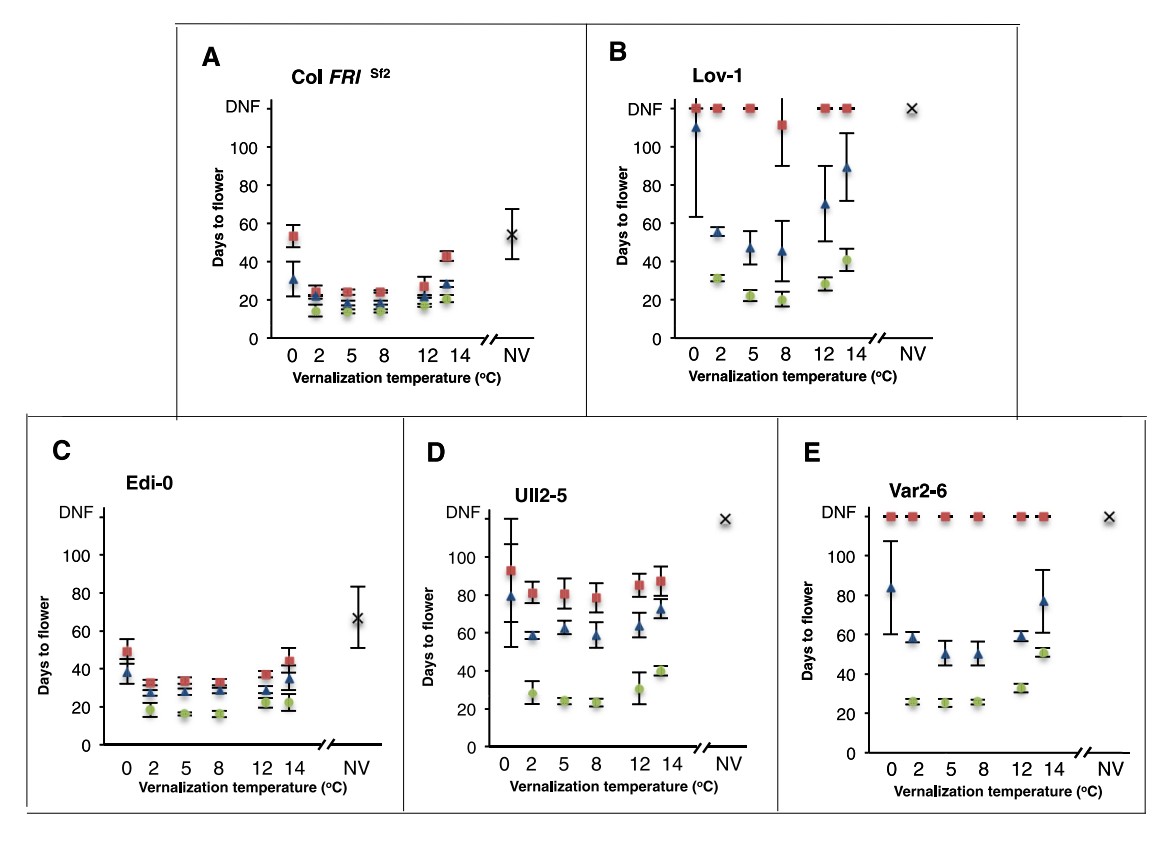

**Figure 1**. Vernalization responses at a range of constant temperatures. Days to flower were recorded for five genotypes after 0 (crosses), 4 (red squares), 6 (blue triangles) and 12 (green circles) weeks vernalization at a range of temperatures, $n \geq 10$. NV = non-vernalized. Error bars = ±S.D.

The following figure supplements are available for figure 1:

**Figure supplement 1**. Map showing accession collection sites.

**Figure supplement 2**. 2 week vernalization of Col $FRI^{Sf2}$ does not reveal differential response to 2 and 5°C treatments.

In order to link the flowering time changes with the changed epigenetic silencing at *FLC* we included a near isogenic line carrying the Lov-1 *FLC* allele (NIL[Lov-1]) in the genetic background of Col $FRI^{Sf2}$ in the field experiments. This line was generated through six generations of introgression and had been genotyped with markers to define the introgressed region (*Figure 3—figure supplement 7*). NIL[Lov-1] took longer to flower than Col $FRI^{Sf2}$ after the first two transfers in 2012 (*Figure 3—figure supplement 5B*). This revealed the clear contribution of the Lov-1 *FLC* allele to differential vernalization response under field conditions, which likely involves the four non-coding polymorphisms in *FLC* close to the nucleation site of the PHD-PRC2 previously defined as underpinning the molecular variation in *FLC* epigenetic silencing between Lov-1 and Col $FRI^{Sf2}$ (*Coustham et al., 2012*).

Expression analysis in the perennial species *Arabidopsis halleri* growing under natural field conditions has shown that plants average temperature over long-term scales (*Aikawa et al., 2010*). It was therefore interesting that the optimal vernalizing temperature for Lov-1 matched the average temperature over the 3-month season when vernalization occurred (*Figure 3B*). We therefore compared vernalization response in the different transplant intervals with vernalization in constant temperatures equivalent to the average temperature of the field conditions (*Figure 4—source data 1*). The different genotypes showed temporal differences in vernalization responsiveness over the three transplant periods in the field. Remarkably, vernalization responsiveness was very similar when the average field temperature was given as a constant temperature, with each genotype showing a different overall profile (*Figure 4*). Indeed, the match is remarkable given the daily oscillations in

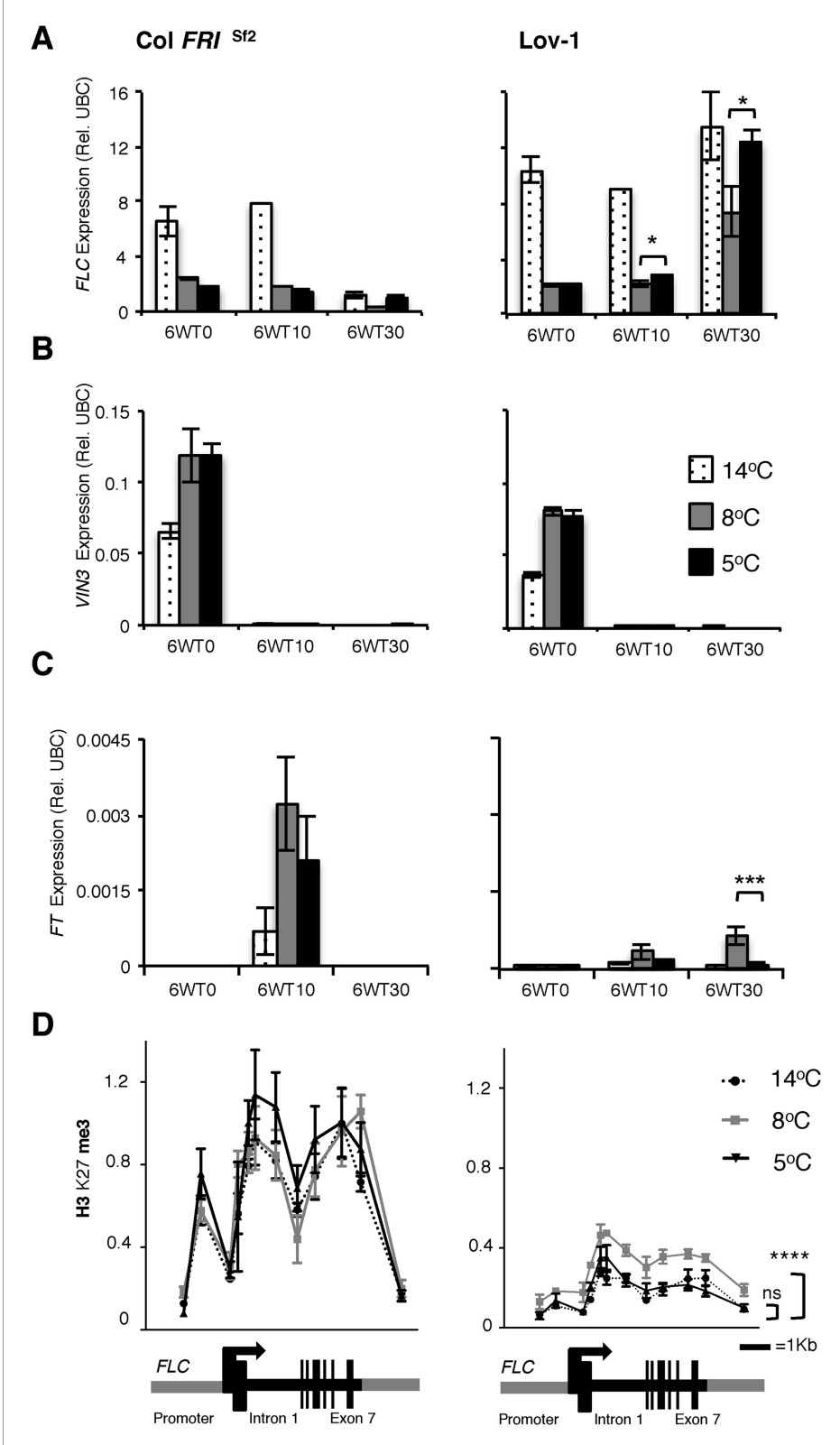

**Figure 2**. Quantitative PCR and ChIP analyses of plants vernalized at 5°C, 8°C and 14°C. Changes in *FLC* (**A**), *VIN3* (**B**) and (**C**) *FT* expression were determined directly after 6 weeks of cold exposure (T0) and again after 10 (T10) and 30 (T30) days subsequent growth at 20°C. Two-tailed Student's *t*-test results: *$p < 0.05$, ***$p < 0.005$. $n = 3$. Error bars = ±S.D. (**D**) H3K27me3 levels over the *FLC* locus were higher for Lov-1 after 6 weeks vernalization at 8°C

*Figure 2. continued on next page*

*Figure 2. Continued*

than 14°C or 5°C (samples were harvested 30 days post cold). **** $p < 0.0001$, Wilcoxon matched-pairs signed rank test on measurements for 12 primer pairs. Error bars = ±S.E.M. NV = nonvernalized, DNF = did not flower and ns = not significant.

The following source data and figure supplement are available for figure 2:

**Source data 1**. Primers used for qPCR ChIP.

**Figure supplement 1**. *FLC* expression determined after 4 weeks of vernalization at a range of temperatures.

temperature especially in the transplant 1 period (*Figure 3A*, *Figure 3—figure supplement 5A*). How plants integrate these fluctuating temperatures over such long timescales is an important area for future molecular dissection.

All the genotypes analysed were found to vernalize effectively during autumn (*Figure 3B*, *Figure 3—figure supplement 5B*), however they have been shown to differ in their seed dormancy (Atwell et al., 2010); accessions from N. Sweden generally have much lower seed dormancy requirement than those from further south (*Debieu et al., 2013*). Thus, the low seed dormancy of Lov-1 would enable germination to occur early enough to exploit the whole of the N. Swedish autumn for vernalization. The increased seed dormancy of S. Swedish accessions (e.g., Ull2-5) is likely to delay germination leading to the necessity for vernalization in some years to extend into winter in S. Sweden. It is interesting to speculate that the reduced effectiveness of temperatures below 5°C for other Swedish accessions Lov-1 and Var2-6 (*Figure 1B,E*) could prevent premature vernalization occurring during unseasonal cool periods in early autumn. Our data also show that the (0°C, 6°C) temperature range widely used to estimate vernalization in *A. thaliana* (*Wilczek et al., 2009*) would only predict partial vernalization of later flowering accessions during our field experiments (*Figure 4—figure supplement 1*). Our data suggest that raising the upper threshold temperature to 15°C would improve estimates of vernalization progress for later flowering accessions under natural field conditions.

In summary, we have employed a combination of molecular and ecological approaches to connect temperature-induced molecular changes at *FLC* with ecologically significant effects in the field. We show that growth at the northern limit of the *A. thaliana* species range has involved a seasonal shift in the timing of vernalization. Perhaps as a response to selection in these extreme conditions one N. Swedish accession, Lov-1, shows a more distinct vernalization temperature optimum that matches the average historical temperature for August-November in that geographical region (*Figure 1*, *Figure 2*, *Figure 3—figure supplement 3A*). Early germination enables vernalization to complete before snowfall and allows flowering to occur directly after snowmelt when the photoperiod and ambient temperatures increase. Rises in global temperature have already reduced vernalization periods to an extent that has impacted the phenology of a range of plant species (*Fitter and Fitter, 2002*; *Cook et al., 2012*). Studies such as this are therefore important to understand how rapidly populations might adapt under future climate scenarios.

## Materials and methods

### Plant material and growth conditions

Genotypes used, standard growth and vernalization conditions have been described previously (*Shindo et al., 2006*). Briefly, plants were sown in a randomized design and stratified for 3 days at 4°C. Seedlings were grown for 7 days at 22°C and then vernalized in cabinets at 14°C, 12°C, 10°C, 8°C (all in Sanyo (Moriguchi, Japan) MLR-351H cabinets), 5°C (walk-in vernalization room), 2°C (modified Liebherr (Kirchdorf, Germany) KP3120) or 0°C (Johnson Controls, Milwaukee, WI). All temperatures were recorded as $\pm \leq 1.5$°C, 70% $\pm \leq 10\%$ RH. An 8hr photoperiod was provided by fluorescent tubes for temperatures $\geq 8$°C and LEDs for temperatures $\leq 2$°C. Plants were transferred to random locations in a controlled environment room (16 hr light, 22°C $\pm$ 2°C) and flowering time was scored as the number of days of growth until floral buds became visible.

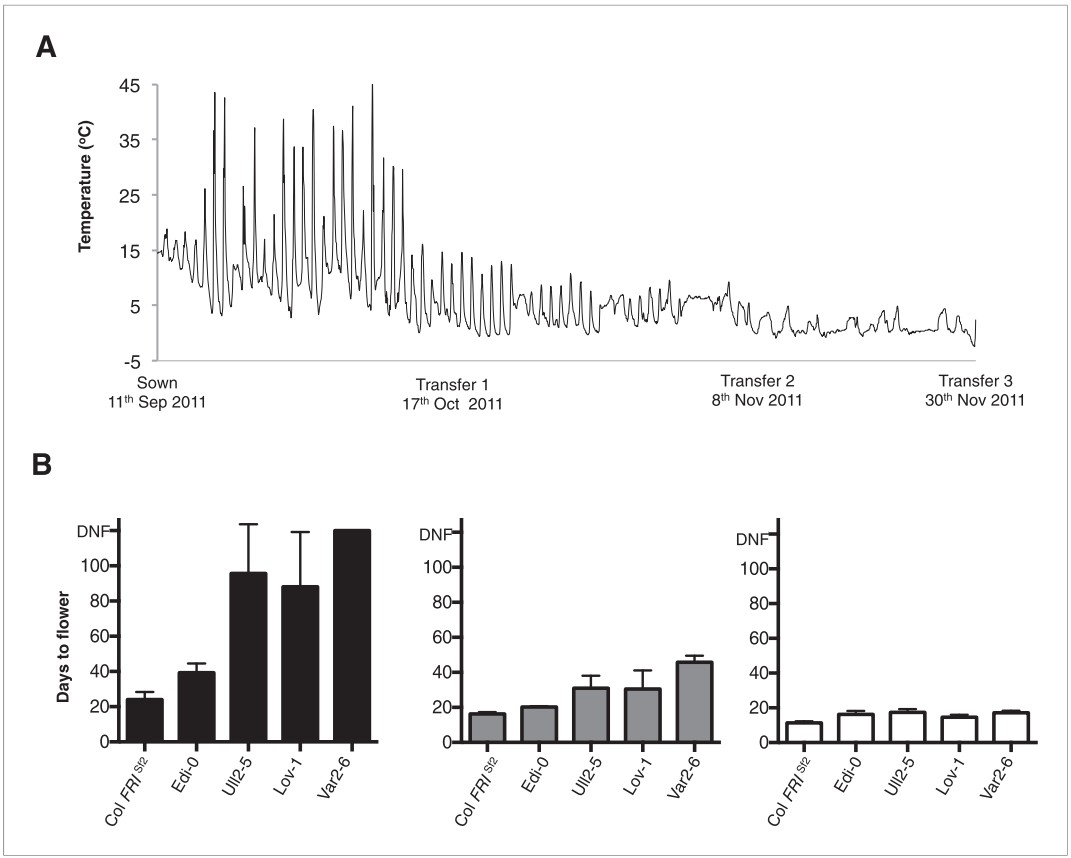

**Figure 3**. Field experiments reveal vernalization occurs in autumn in northern Sweden. (**A**) Date of sowing and plant transfers to the greenhouse are shown with hourly soil surface temperatures recorded during autumn 2011. (**B**) Days to flower recorded after plants were transferred to a warmed greenhouse at three time points during autumn: Transfer 1 (black), Transfer 2 (grey) and Transfer 3 (white). $n \geq 10$. Error bars = ±S.D.

The following source data and figure supplements are available for figure 3:

**Source data 1**. Developmental stage of natural *Arabidopsis thaliana* populations in spring in the High Coast area of N. Sweden (62.5°N).

**Figure supplement 1**. The Lov-1 natural population flowers rapidly after snowmelt in spring.

**Figure supplement 2**. Snow consistently covers and protects plants from subzero air temperatures during winter.

**Figure supplement 3**. Temperature records from N. Sweden near Lövvik.

**Figure supplement 4**. Field locations and climate data collection sites in Sweden.

**Figure supplement 5**. Sweden field experiments results 2012.

**Figure supplement 6**. Plants flowered synchronously with natural populations after 5 months of continuous snow cover.

**Figure supplement 7**. Genetic map showing Lov-1 introgressed region on chromosome 5.

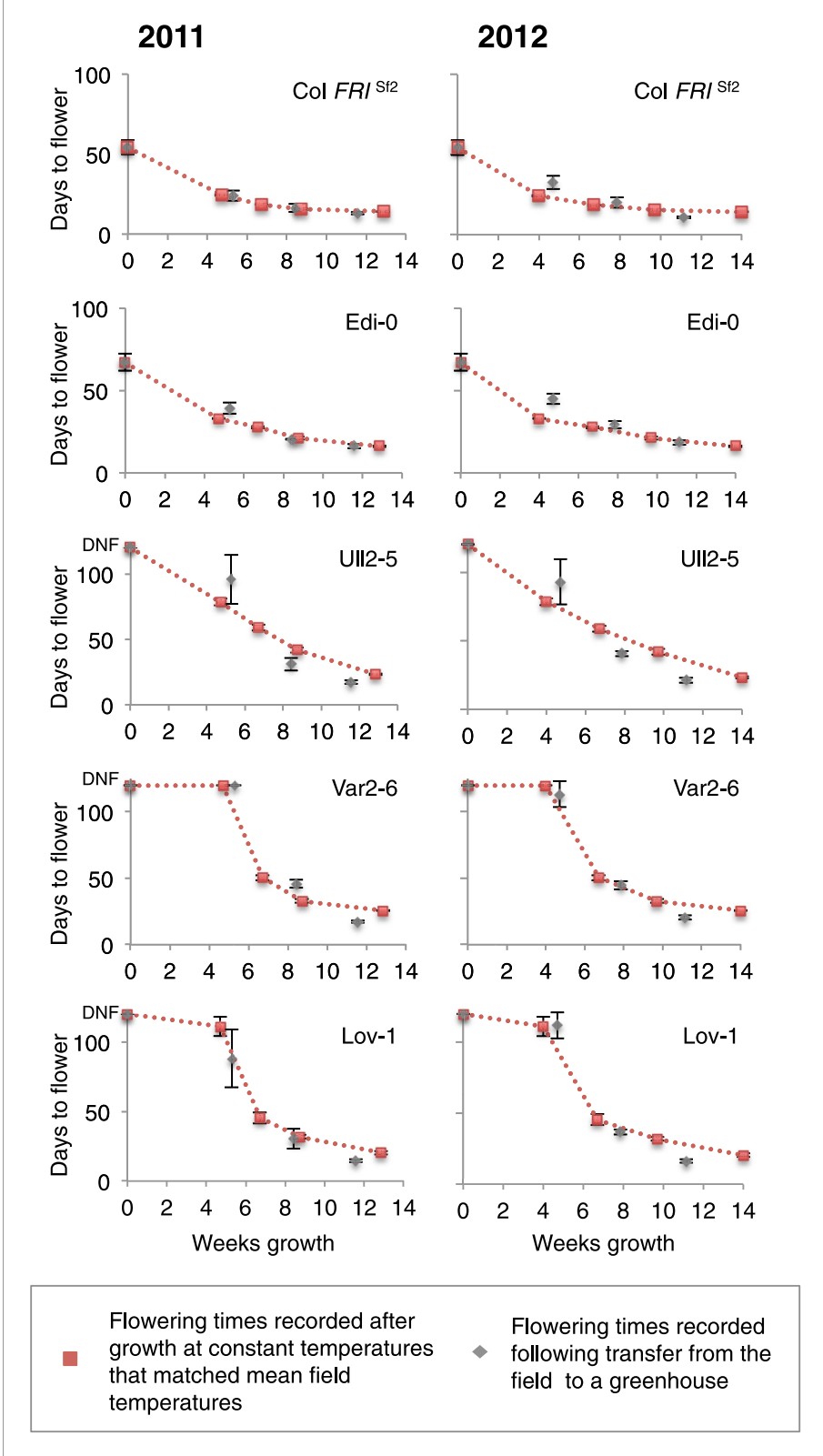

**Figure 4**. Prediction of vernalization response under field conditions. Days to flower recorded after the three transplants during field experiments in 2011 and 2012 are shown in grey. Red dashed lines indicate changes in

*Figure 4. continued on next page*

*Figure 4. Continued*

flowering time estimated by flowering time results observed after vernalization at constant temperatures. Error bars represent ±S.D. $n \geq 10$, DNF = did not flower.

The following source data and figure supplement are available for figure 4:

**Source data 1**. Cabinet flowering time data were selected where conditions most closely matched mean temperatures recorded during 2011 and 2012 field experiments.

**Figure supplement 1**. Accumulation of temperatures within different effective vernalization ranges.

## Expression analysis

Total RNA was extracted as described previously (*Box et al., 2011*). cDNA was synthesized using Precision nano-script reverse transcription (Primerdesign) with oligo d(T) and analysed by qPCR on a LightCycler 480 II intrument (Roche, Basel, Switzerland), using LightCycler 480 Probes Master mix (Roche). *FLC* mRNA was assayed using Roche Universal Probe Library (UPL) #65 (5′-ctggagga-3′) with primers sFLC_UPL_F (5′-gtgggatcaaatgtcaaaaatg-3′) and sFLC_UPL_R (5′-ggagagggcagtctcaaggt-3′). *VIN3* mRNA was assayed using UPL#67 (5′-tggtggat-3′) with primers VIN3_UPL_F (5′-cgcgtattgcggtaaagataa-3′) and VIN3_UPL_R (5′-tctctttcgccaccttcact-3′). *FT* mRNA was assayed using UPL#138 (5′-tggtggat-3′) with primers FT_UPL_#138_F (5′-ggtggagaagacctcaggaa-3′) and FT_UPL_#138_R (5′-ggttgctaggacttggaacatc-3′). Expression of each gene was normalized to *UBC* (*At5g25760*) with primers UBC_UPL_F (5′-tcctcttaactgcgactcagg-3), UBC_UPL_R (5′-gcgaggcgtgtatacatttg-3) and UPL#9 (5′-tggtgatg-3′). Statistical analyses of logged expression data were performed using GraphPad Prism version 6 software (La Jolla, CA).

## ChIP and real-time quantitative PCR analysis

ChIP assays were performed as previously described (*Sun et al., 2013*) using H3K27me3 and H3 antibodies cited by *Angel et al. (2011)*. Primers used in this analysis are shown in *Figure 2—source data 1*. *SHOOT MERISTEMLESS* (*STM*) was used as the internal control and data are represented as the ratio of (H3K27me3*FLC*/H3 *FLC*) to (H3K27me3 *STM*/H3 *STM*). Statistical analysis of ChIP data was performed using GraphPad Prism version 5 software for Mac.

## Climate analysis

Hourly temperatures were recorded using Tinytag data-loggers (Chichester, UK). Historical climate data were obtained from Swedish Meteorological and Hydrological Institute. Three temperature and snow-depth readings taken at 0600 hr, 1200 hr and 1800 hr were used to calculate daily means. Boxplots graphs were created using QI Macros add-ins for Excel (Denver, CO). Statistical analyses of climate data were performed using GraphPad Prism version 6 software.

## Field experiments

Seeds were stratified for 4 days at 5℃, sown into trays using a randomized block design and placed outside (62° 23.463′N, 17° 18.272′E). Seedlings were thinned to one plant per cell after 7 days and then transferred to Ramsta (62° 50.988′N, 18° 11.570′E) 1 week later. At each transfer date, plants were returned to a greenhouse in Mid-Sweden University, Sundsvall (16 hr light, 22℃ ± 2℃) where flowering time was determined as the number of days growth until floral buds became visible.

## Acknowledgements

The Dean lab is supported by the UK Biotechnology and Biological Sciences Research Council (BBSRC) Institute Strategic Programme grant BB/C517633/1 and a European Research Council Advanced Investigator grant 233039 ENVGENE. SD is supported by a studentship from the Earth and Life Systems Alliance, a joint venture between the John Innes Centre and the University of East Anglia. We thank Arthur Korte at Gregor Mendel Institute, Vienna for assaying the vernalization responses at 0℃, Öhmans farm for field site and all the members of the Dean lab for useful discussions.

# Additional information

### Funding

The funders had no role in study design, data collection and interpretation, or the decision to submit the work for publication.

### Author contributions

SD, Conception and design, Acquisition of data, Analysis and interpretation of data, Drafting or revising the article; SH, Designed the field experiments, Acquisition of data, Drafting or revising the article; JQ, Acquisition of data, Drafting or revising the article; JI, Analysis and interpretation of data, Drafting or revising the article; AG, Conception and design, Drafting or revising the article; CD, Conception and design, Analysis and interpretation of data, Drafting or revising the article

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
