## [Decision Letter]

Thank you for choosing to send your work entitled “Seasonal shift in timing of vernalization as an adaptation to extreme winter” for consideration at *eLife*. Your full submission has been evaluated by Detlef Weigel (Senior editor and peer reviewer) and one other peer reviewer, and the decision was reached after discussions between the two reviewers. Based on our discussions, we regret to inform you that your work will not be considered further for publication in *eLife*.

The manuscript was seen by two reviewers, one of whom had reviewed the previous version. The work addresses the question whether accessions of *Arabidopsis thaliana* complete vernalization at different times relative to the height of winter, as a consequence of adaptation to geographic differences in winter temperatures and winter duration. A model presented in many reviews is that vernalization equates exposure to winter-like temperature, and thus is completed only in spring. It is, however, unclear how rigorously this has been tested before. A previous paper by Wollenberg and Amasino reported that flowering of some Spanish accessions is accelerated by transient exposure to quite high vernalization temperatures, and Méndez-Vigo et al. have hypothesized that “mild and moderately cold winters activate the vernalization pathway to promote flowering in winter” (of Spanish accessions).

Both reviewers agreed that there are two major conclusions from the present manuscript one can be confident about: (i) Vernalization of Lov-1 is essentially completed in late fall, prior to cover by snow, such that the plants are already competent to flower as soon as snow melts later in the spring, and (ii) the optimal vernalization temperature for Lov-1 seems to be in the 5-8 degree range.

Based on logic alone, it seems likely that this vernalization behavior has to be common in places with regular and consistent snow cover. An interesting question is how vernalization behavior differs in other regions. Knowledge of this is essential in order to support the claim that there is a “seasonal shift in timing of vernalization as an adaptation to extreme winter”. Figure 3 seems to nicely demonstrate that a few other accessions are also vernalized when sown in fall and transferred to a greenhouse in November. To conclusively answer the question that you set out to answer, one would, however, need to conduct similar experiments at other locales, since it is not clear when *Arabidopsis* is vernalized at more Southern locations.

Both reviewers had also similar general concerns regarding the role of FLC in the observed behavior:

1) The behavior of near isogenic introgression lines indicated that the FLC locus contributes to this flowering behavior, but there are clearly additional factors, as the FLC region even in conjunction with another linked region on chr 5 has a much weaker effect on flowering than what was observed with Lov-1 itself. The conclusion “that the changed vernalization temperature response if Lov-1 involves a complex interaction between FLC and gene products at linked loci” is certainly not incorrect, but it does not provide the whole story.

2) Following on from this line of arguments, why did you use NIL-1, which contains the undesired downstream region, in your field experiments, rather than NIL-2, which apparently only contains FLC from Lov-1? This would seem to complicate the interpretation of the genetic evidence, especially because this downstream region appears to contribute a 13-20 day delay in flowering time. Depending on which NIL is used as the denominator, this is either a 30% or a 60% delay. Your comments on this region in your presentation of the results were not entirely convincing, and the choice of NIL-1 for the field experiment appears somewhat unfortunate.

[Editors’ note: a previous version of this study was rejected after peer review, but the authors submitted for reconsideration. The previous decision letter after peer review is shown below.]

Thank you for choosing to send your work entitled “Seasonal shift in timing of vernalization as an adaptation to extreme winter” for consideration at *eLife*. Your full submission has been evaluated by Detlef Weigel (Senior editor) and two other peer reviewers, and the decision was reached after discussions between the reviewers. We regret to inform you that your work will not be considered further for publication.

You have studied the potential ecological significance of a northern Swedish allele (Lov-1) of the FLC gene of Arabidopsis thaliana by integrating molecular genetics, field observations, meteorological data and field-greenhouse ecological experiments. A surprising observation is that vernalization is already completed during fall at a rather higher temperature of about 8°C, which enables very rapid flowering opening after snow melt. This is a potentially exciting new insight because vernalization has been typically considered to occur during winter to prevent early flowering. What is lacking, however, is a demonstration that these prior assumptions were justified, and that the Lov-1 allele has indeed a different vernalization temperature optimum in addition to its different vernalization length requirement. What impressed the reviewers most was the very clever idea to take plants grown at natural field sites into the greenhouse, in order to test their vernalization status. This nicely confirms that Lov-1 needs longer vernalization than the reference strain, although this is perhaps difficult to interpret because the reference strain flowers so much more quickly anyway.

From a series of elegant experiments in your lab, it was known that natural lines of *Arabidopsis thaliana* differ in their requirements for length of vernalization treatment, and that much of that variation maps to the FLC gene itself. In this paper, you have asked whether there are also different temperature optima for vernalization. In Figure 1, you contrast a line with the FLC standard allele, Col FRI-Sf2, with the Lov-1 line. You find that in both lines 2, 5, or 8°C are more effective for vernalization than temperatures below or above. The critical question is whether there are differences between the two lines at 12 and 14°C. Because the effect of vernalization on flowering of the reference line is much smaller than for Lov-1, the reviewers were unconvinced of this claim. Not only was there no proper statistical treatment, but it is even unclear whether this can be resolved by statistics, because it would be rather arbitrary how one defines vernalization response, i.e., as absolute difference in flowering, relative acceleration of flowering etc. More importantly, the reviewers agreed that these experiments mostly confirmed that Lov-1 needs to be vernalized much longer than Col FRI-Sf2 rather than that the results demonstrated a different temperature optimum. That there is perhaps less of a difference between the different temperature regimens than claimed can also be deduced from Figure 1. If one of the lines has a temperature optimum at 2-8°C, it would seem to be Col FRI-Sf2, not Lov-1, where 12°C seems to be more effective in reducing FLC expression than in Col FRI-Sf2. The H3K27me3 data in Figure 1 are more in support of differential temperature effects, but the differences between 5 and 8°C for Lov-1 are not reflected in obvious differences in flowering behavior, so the results are equivocal.

In the end, the reviewers remained unconvinced that there is evidence for “a changed temperature integration mechanism involving FLC epigenetic silencing (that) has led to a phenological shift in vernalization timing”. For such a conclusion, one would need to have seen similar experiments at other sites, with lines that supposedly have different temperature optima (and preferentially similar vernalization length requirement, because it seems that length and temperature effects are difficult to disentangle).

Major issues of the manuscript to be addressed include:

1) There are a few issues regarding the paragraph: “Hourly air temperature data collected near Lövvik from 1st August until snow cover in 2008, 2009 and 2013 revealed mean average autumn temperatures close to the 8°C; the optimal Lov-1 constant vernalization temperature (Figure 2). National climate records showed the longer-term (1961 to 2008) average daily temperature in that seasonal period was 8.86°C (SD = 0.63) with 86% of average daily temperatures shown in Figure 2 within the range identified to be effective for vernalization of Lov-1 seedlings. Together these data raised the possibility that the change in thermal sensitivity of vernalization was an adaptation beneficial for reproductive success in northern Sweden.”

1-1) Why was the average started from 1st August? Depends on the day of start, the average until snow cover can be arbitrary changed. I guess authors could show ecological argument or data, such as the germination timing in the locality.

1-2) Why does mean average temperature matters? While the lab experiment was conducted in constant 8°C, the natural temperature would fluctuate in each day. Rather than daily average, it is possible that the coldest temperature may matter, or hours lower than a certain threshold temperature may matter (model by [1]), or whatever. I would propose either (1) defend the usage of daily average temperature (2) show experimental data supporting the significance daily temperature fluctuation, or (3) remove the argument (importantly abstract) of the coincidence between lab experiment and natural condition, although it would weaken the paper.

1-3) Describe the detailed data of the national climate records. Does it provide only average daily temperature? Even if hourly temperature is not available, maximum and minimum temperatures may be relevant.

2) The second last paragraph of Results and Discussion on modeling as well as Figure 4–table supplement 1: Too few details are described about temperature both in experiments and models. After all, in which condition were the plants grown? The “Plant material and growth conditions” section of the Methods does not seem to describe the conditions of the plants described in the table, which were grown in 12, 10, 8, 5°C. Moreover, nothing is written on the model in the Materials and methods. Although there is a short description in the figure legend, it is far from adequate.

3) In the field experiments (Figure 3), the NIL1Lov-1 and Lov-1 behaves similarly, but in the laboratory experiments (Figure 3—figure supplement 5), they seem very different in the days to flowering. Please explain this point.

4) Regarding the second last sentence of the second last paragraph “Rapid flowering in spring may be required to complete reproductive development in the relatively short northern latitude summer”, please discuss the reasoning more. It would depend on the germination timing. Suppose they flower in May, make seeds in June, germinates in August, is rapid flowering really advantageous? Other possibilities may be discussed. The temperature may become unfavorable quickly in spring, or herbivores may favor early flowering (e.g. a study of *A. thaliana* in Sweden, Akiyama and Agren, Conflicting selection on the timing of germination in a natural population of *Arabidopsis thaliana*. J Evol Biol. 2013; Akiyama and Agren, Magnitude and timing of leaf damage affect seed production in a natural population of *Arabidopsis thaliana* (Brassicaceae). PLoS One. 2012;7(1):e30015; a study using *A. halleri*, Kawagoe and Kudoh, Escape from floral herbivory by early flowering in *Arabidopsis halleri* subsp. gemmifera. Oecologia. 2010 Nov;164(3):713-20).

---

## [Author Response]

*[…] Both reviewers had also similar general concerns regarding the role of FLC in the observed behavior*:

*1) The behavior of near isogenic introgression lines indicated that the FLC locus contributes to this flowering behavior, but there are clearly additional factors, as the FLC region even in conjunction with another linked region on chr 5 has a much weaker effect on flowering than what was observed with Lov-1 itself. The conclusion “that the changed vernalization temperature response if Lov-1 involves a complex interaction between FLC and gene products at linked loci” is certainly not incorrect, but it does not provide the whole story*.

*2) Following on from this line of arguments, why did you use NIL-1, which contains the undesired downstream region, in your field experiments, rather than NIL-2, which apparently only contains FLC from Lov-1? This would seem to complicate the interpretation of the genetic evidence, especially because this downstream region appears to contribute a 13-20 day delay in flowering time. Depending on which NIL is used as the denominator, this is either a 30% or a 60% delay. Your comments on this region in your presentation of the results were not entirely convincing, and the choice of NIL-1 for the field experiment appears somewhat unfortunate*.

In light of the reviewer’s comments we have undertaken field studies to establish a lack spring germination in Northern Sweden, and included data from additional genotypes to show that despite different vernalization temperature profiles, all accessions vernalize during autumn in Northern Sweden.

This reinforces our central conclusion that there is a shift in the timing of vernalization as an adaptation to extreme winters. Our observation of vernalization between 0^°^C and 14^°^C challenges current dogma and opens up the opportunity for complete vernalization during autumn.

The specific changes are:

Figure 1: Var2-6 flowering time data has been added. Expression and ChIP analyses have now been moved to Figure 2 and four week *FLC* expression graphs are now included as Figure 2—figure supplement 1.

Figure 1—figure supplement 1: A new map has been included that shows collection sites of the natural accessions.

Figure 2: Climate data analyses have been moved to Figure 3—figure supplement 2 and Figure 3—figure supplement 3 and has been replaced by molecular data in Figure 2. For completeness we now include expression data for the 6 week treated samples alongside the ChIP data. This shows significantly lower levels of *FLC* reactivation and higher *FT* induction following an 8^°^C treatment versus a 5^°^C treatment despite similar levels of *VIN3* induction observed for both temperatures.

Figure 3: Var2-6 data has now been added to the 2011 field results.

Figure 3–table supplement 1 has been added. This describes field studies we have undertaken to analyse spring germination in Northern Sweden.

Field experiment results (2011: Figure 3; 2012: Figure 3—figure supplement 5) now include Var2-6 flowering data. This demonstrates that autumn vernalization also occurs for a representative accession from the Northern Swedish *FLC* haplotype group.

2012 field results (Figure 3—figure supplement 5) now include data for the introgressed line that contains only the Lov-1 *FLC* region. References to the other NILs (referred to as NIL1^Lov-1^ and NIL3^Lov-1^ in our previous submission) have now been removed from the manuscript.

Figure 3—figure supplement 5 now includes Var2-6 flowering time data.

Figure 4 has been reconfigured to include Var2-6 field and constant temperature responses.

[Editors’ note: the author responses to the previous round of peer review follow.]

Thank you for the very constructive comments on our previous submission. We realized that in our attempt to simplify the story we chose not to include results that in hindsight would have helped convince the reviewers of our conclusions. We have therefore rewritten the manuscript including all these additional data.

*You have studied the potential ecological significance of a northern Swedish allele (Lov-1) of the FLC gene of Arabidopsis thaliana by integrating molecular genetics, field observations, meteorological data and field-greenhouse ecological experiments. A surprising observation is that vernalization is already completed during fall at a rather higher temperature of about 8°C, which enables very rapid flowering opening after snow melt. This is a potentially exciting new insight because vernalization has been typically considered to occur during winter to prevent early flowering*.

We are particularly pleased that the reviewers recognized the importance of this finding. The view that vernalization occurs during winter is engrained in the thinking of the plant community and determines experimental strategy, inputs to modeling phenology and interpretations of field experiments.

*What is lacking, however, is a demonstration that these prior assumptions were justified, and that the Lov-1 allele has indeed a different vernalization temperature optimum in addition to its different vernalization length requirement*.

In our previous submission we had chosen to present two extreme strategies. We realize now that our attempt to keep things simple had led to confusion. In our new submission we now include other accessions and so can clearly show that length of cold period can be separated from the vernalization temperature optimum. We also demonstrate that our prior assumption of winter vernalization can be justified by showing that current phenological model parameters underestimate vernalization progress in natural field conditions.

*What impressed the reviewers most was the very clever idea to take plants grown at natural field sites into the greenhouse, in order to test their vernalization status. This nicely confirms that Lov-1 needs longer vernalization than the reference strain, although this is perhaps difficult to interpret because the reference strain flowers so much more quickly anyway*.

The addition of data from other accessions, particularly Ull2-5, will hopefully convince the reviewers that rapid flowering and temperature optimum are separable.

*From a series of elegant experiments in your lab, it was known that natural lines of* Arabidopsis thaliana *differ in their requirements for length of vernalization treatment, and that much of that variation maps to the FLC gene itself. In this paper, you have asked whether there are also different temperature optima for vernalization. In*
Figure 1*, you contrast a line with the FLC standard allele, Col FRI-Sf2, with the Lov-1 line. You find that in both lines 2, 5, or 8°C are more effective for vernalization than temperatures below or above. The critical question is whether there are differences between the two lines at 12 and 14°C. Because the effect of vernalization on flowering of the reference line is much smaller than for Lov-1, the reviewers were unconvinced of this claim*.

Indeed we found both Swedish accessions vernalize slightly better at 12 than 14^°^C, with Edi-0 and Col *FRI*^Sf2^ vernalizing equally efficiency at both temperatures. However, we do not find a functional significance for this difference in the field experiments, so do not consider it a critical issue. What does appear to influence field behaviour is the differential response between 2 and 5/8^o^C. We have also included experiments on a short vernalization of Col *FRI*^Sf2^ that demonstrate that partial vernalization of this faster flowering line does not reveal a differential temperature effect between 2 and 8^°^C.

*Not only was there no proper statistical treatment, but it is even unclear whether this can be resolved by statistics, because it would be rather arbitrary how one defines vernalization response, i.e., as absolute difference in flowering, relative acceleration of flowering etc. More importantly, the reviewers agreed that these experiments mostly confirmed that Lov-1 needs to be vernalized much longer than Col FRI-Sf2 rather than that the results demonstrated a different temperature optimum. That there is perhaps less of a difference between the different temperature regimens than claimed can also be deduced from*
Figure 1.

We have now used 4 week treated rather than 6 week cold treated plants for the analysis in Figure 1. These show a significant difference in the temperature-dependent re-activation of *FLC* transcription 30 days after transfer to warm. This difference in re-activation does translate into flowering time differences.

*If one of the lines has a temperature optimum at 2-8°C, it would seem to be Col FRI-Sf2, not Lov-1, where 12°C seems to be more effective in reducing FLC expression than in Col FRI-Sf2*.

We have plotted absolute expression values in Figure 1 rather than normalized to NV to demonstrate that 12^°^C is not more effective in Lov-1 than Col *FRI*^Sf2^.

*The H3K27me3 data in*
Figure 1
*are more in support of differential temperature effects, but the differences between 5 and 8°C for Lov-1 are not reflected in obvious differences in flowering behavior, so the results are equivocal*.

We hope that the new Figure 1 convinces the reviewer that the chromatin differences are reflected in changed expression. We have also added the ChIP data at 14^°^C and statistical test results that strengthen the view of an 8^°^C optimum.

*In the end, the reviewers remained unconvinced that there is evidence for “a changed temperature integration mechanism involving FLC epigenetic silencing (that) has led to a phenological shift in vernalization timing”. For such a conclusion, one would need to have seen similar experiments at other sites, with lines that supposedly have different temperature optima (and preferentially similar vernalization length requirement, because it seems that length and temperature effects are difficult to disentangle)*.

We have extensively edited the whole manuscript to more accurately describe our major findings:

- Vernalization in *A. thaliana* accessions, including one from the northern limit of the range, is inefficient at 0^°^C but still effective at 14^°^C;

- Current vernalization threshold parameters in *Arabidopsis* flowering models only predict a partial response for Lov-1 before winter, but we present field results validating our prediction that vernalization completes during autumn in N. Sweden;

- Lov-1 shows that both the effective range and optimal vernalization response match historical N. Sweden average autumn temperatures. The vernalization optima involves cis polymorphism and altered chromatin silencing.

*Major issues of the manuscript to be addressed include*:

*1) There are a few issues regarding the paragraph: “Hourly air temperature data collected near Lövvik from 1st August until snow cover in 2008, 2009 and 2013 revealed mean average autumn temperatures close to the 8°C; the optimal Lov-1 constant vernalization temperature (*Figure 2*). National climate records showed the longer-term (1961 to 2008) average daily temperature in that seasonal period was 8.86°C (SD = 0.63) with 86% of average daily temperatures shown in*
Figure 2
*within the range identified to be effective for vernalization of Lov-1 seedlings. Together these data raised the possibility that the change in thermal sensitivity of vernalization was an adaptation beneficial for reproductive success in northern Sweden*.*”*

*1-1) Why was the average started from 1st August? Depends on the day of start, the average until snow cover can be arbitrary changed. I guess authors could show ecological argument or data, such as the germination timing in the locality*.

The 1^st^ of August was used as the starting date for temperature data as this coincided with the observed germination time.

*1-2) Why does mean average temperature matters? While the lab experiment was conducted in constant 8°C, the natural temperature would fluctuate in each day. Rather than daily average, it is possible that the coldest temperature may matter, or hours lower than a certain threshold temperature may matter (model by*
[1]*), or whatever. I would propose either (1) defend the usage of daily average temperature (2) show experimental data supporting the significance daily temperature fluctuation, or (3) remove the argument (importantly abstract) of the coincidence between lab experiment and natural condition, although it would weaken the paper*.

We looked at average daily temperatures because this is a common method used in agriculture to determine the minimum quantity of cold required by a crop to ensure synchronous flowering and maximum yield. We also considered long-term average temperature because a long 6 week integration period was reported to contribute to the epigenetic repression of *A. halleri* under field conditions (1). As with many crops, the observed vernalization response of *A. thaliana* does not correlate well with daily maximum or minimum temperatures.

*1-3) Describe the detailed data of the national climate records. Does it provide only average daily temperature? Even if hourly temperature is not available, maximum and minimum temperatures may be relevant*.

Clearer descriptions of the climate data and the analysis have now been included in the Methods section. But briefly, average temperatures were calculated from national records of three daily readings taken at 0600hrs, 1200hrs and 1800hrs.

*2) The second last paragraph of Results and Discussion on modeling as well as Figure 4–table supplement 1: Too few details are described about temperature both in experiments and models. After all, in which condition were the plants grown? The “Plant material and growth conditions” section of the Methods does not seem to describe the conditions of the plants described in the table, which were grown in 12, 10, 8, 5°C. Moreover, nothing is written on the model in the Materials and methods. Although there is a short description in the figure legend, it is far from adequate*.

We have now included thorough descriptions of growth conditions and altered the layout of Figure 4–table supplement 1 to make the cabinet data selection process clearer.

*3) In the field experiments (*Figure 3*), the NIL1Lov-1 and Lov-1 behaves similarly, but in the laboratory experiments (*Figure 3—figure supplement 5*), they seem very different in the days to flowering. Please explain this point*.

Indeed the NIL1 Lov-1 plants were much later in the field than after the equivalent treatment in the chamber, presumably reflecting important environmental difference still to be fully determined. This observation has now been included in our new submission.

*4) Regarding the second last sentence of the second last paragraph “Rapid flowering in spring may be required to complete reproductive development in the relatively short northern latitude summer”, please discuss the reasoning more. It would depend on the germination timing. Suppose they flower in May, make seeds in June, germinates in August, is rapid flowering really advantageous? Other possibilities may be discussed. The temperature may become unfavorable quickly in spring, or herbivores may favor early flowering (e.g. a study of* A. thaliana *in Sweden, Akiyama and Agren, Conflicting selection on the timing of germination in a natural population of* Arabidopsis thaliana*. J Evol Biol. 2013; Akiyama and Agren, Magnitude and timing of leaf damage affect seed production in a natural population of* Arabidopsis thaliana *(Brassicaceae). PLoS One. 2012;7(1):e30015; a study using* A. halleri*, Kawagoe and Kudoh, Escape from floral herbivory by early flowering in* Arabidopsis halleri *subsp. gemmifera. Oecologia. 2010 Nov;164(3):713-20)*.

We have now extended the Discussion to include both observed germination times and experimental evidence for a lack of dormancy in addition to the potential advantages of early flowering for this accession.